# Real-World Analysis of the Clinical and Economic Impact of the 21-Gene Recurrence Score (RS) in Invasive Lobular Early-Stage Breast Carcinoma in Ireland

Lynda M. McSorley [1,*], Mehala Tharmabala [1], Fathiya Al Rahbi [2], Fergus Keane [1], Denis Evoy [3], James G. Geraghty [3], Jane Rothwell [3], Damian P. McCartan [3], Megan Greally [4], Miriam O'Connor [5], Deirdre O'Mahony [6], Maccon Keane [7], Michael John Kennedy [8], Seamus O'Reilly [9], Steve J. Millen [10], John P. Crown [1], Catherine M. Kelly [11], Ruth S. Prichard [3], Cecily M. Quinn [2,12] and Janice M. Walshe [1,12]

1   Department of Medical Oncology, St. Vincent's University Hospital, D04 T6F4 Dublin, Ireland
2   Department of Pathology, St. Vincent's University Hospital, D04 T6F4 Dublin, Ireland
3   Department of Surgery, St. Vincent's University Hospital, D04 T6F4 Dublin, Ireland
4   Department of Medical Oncology, Beaumont Hospital, D04 T6F4 Dublin, Ireland
5   Department of Medical Oncology, University Hospital Waterford, X91 ER8E Waterford, Ireland
6   Department of Medical Oncology, Bon Secours Hospital, T12 DV56 Cork, Ireland
7   Department of Medical Oncology, Galway University Hospitals, H91 YR71 Galway, Ireland
8   Department of Medical Oncology, St James's Hospital, D08 NHY1 Dublin, Ireland
9   Department of Medical Oncology, Cork University Hospital, T12 DC4A Cork, Ireland
10  Exact Sciences UK Ltd., London EC4M 9AF, UK
11  Department of Medical Oncology, The Mater Misericordiae University Hospital, D07 R2WY Dublin, Ireland
12  School of Medicine, University College Dublin, D04 V1W8 Dublin, Ireland
*   Correspondence: lyndamcsorley@svhg.ie; Tel.: +353-1-221-4000

**Abstract: Background:** This study, using real-world data, assesses the impact of RS testing on treatment pathways and the associated economic consequences of such testing. This paper pertains to lobular breast cancer. **Methods:** A retrospective, observational study was undertaken between 2011 and 2019 on a cross-section of hormone receptor-positive (HR+), HER2-negative, lymph node-negative, early-stage breast cancer patients. All patients had ILC and had RS testing in Ireland. The patient population is representative of the national population. Patients were classified as low (RS ≤ 25) or high (RS > 25) risk. Patients aged ≤50 were stratified as low (RS 0–15), intermediate (RS 16–25), or high risk (RS > 25). **Results:** A total of 168 patients were included, most of whom had grade 2 (G2) tumors (*n* = 154, 92%). Overall, 155 patients (92.3%) had low RS (≤25), 12 (7.1%) had high RS (>25), and 1 (0.6%) had unknown RS status. In 29 (17.5%) patients aged ≤50 at diagnosis, RS was ≤15 in 16 (55%), 16–20 in 6 (21%), 21–25 in 5 (17%), >25 in 1 (3.5%), and unknown in 1 (3.5%). Post RS testing, 126 patients (78%) had a change in chemotherapy recommendation; all to hormone therapy. In total, only 35 patients (22%) received chemotherapy. RS testing achieved a 75% reduction in chemotherapy use, resulting in savings of €921,543.84 in treatment costs, and net savings of €387,283.84. **Conclusions:** The use of this test resulted in a 75% reduction in chemotherapy and a significant cost savings in our publicly funded health system.

**Keywords:** breast cancer; gene expression profiling; prognostic factors; cost analysis

## 1. Introduction

Breast cancer remains the most commonly occurring cancer in women worldwide and poses a significant health challenge across many healthcare settings. In Ireland, there are over 3000 new cases of breast cancer diagnosed annually, and one in nine women will develop breast cancer in their lifetime [1]. Early-stage breast cancer accounts for 70% of newly diagnosed breast cancer cases in Ireland, and although survival rates are high (82% 5-year survival), it remains the second most common cause of cancer death in women [1].

Breast cancer is characterised by heterogenous tumours that display distinct histological and molecular features. Approximately 80% of hormone receptor-positive breast cancers are invasive ductal carcinoma (IDC), while the remaining 10–15% are invasive lobular carcinoma (ILC). Over the last two decades, in particular, there has been a significant increase in the incidence of ILC globally [2]. Furthermore, ILC is arguably more than a histologic variant; as a distinct subset of breast cancer, it presents unique management challenges and prognostic implications.

ILC often has many initially favourable prognostic factors, including hormone receptor positivity and a lower tumour grade, as well as lower proliferation rates than IDC. However, long-term survival data suggest that ILC loses that early advantage at approximately 10 years, with late recurrence seen, the likelihood of which increases with grade [3]. Equally relevant is the observation that adjuvant chemotherapy does not appear to equate to better cancer-specific survival in ILC [2,4]. Adding genomic data to clinicopathological information could better inform adjuvant treatment decisions, where data already suggest that adjuvant chemotherapy has limited benefit. A rationalised approach to patient selection for genomic testing has the potential to improve resource utilisation and patient satisfaction; for example, in patients with low-grade ILC, other factors such as Ki67 and PR status may offer sufficient prognostic information to render the use of genomic testing unnecessary in this subgroup [5]. By better selecting those patients likely to benefit from chemotherapy, fewer patients will be exposed to chemotherapy and the short- and long-term toxicities of cytotoxic treatment, such as nausea, alopecia, early menopause, infertility, and the psychological impact of these effects.

The 21-gene test recurrence score (RS) (Oncotype Dx, Genomic Health, Redwood City, CA, USA) is a reverse transcriptase polymerase chain reaction assay that measures the quantitative expression of specific mRNA for 16 cancer genes and 5 reference genes in a breast cancer tumour block. The resulting score, ranging from 0 (low risk) to 100 (high risk), is used to estimate the risk of recurrence over time in patients with HR+, HER2-negative breast cancer [6,7]. With redefined TAILOR-X trial data, updated parameters are presented: low risk (RS 0–15), intermediate risk (RS 16–25), and high risk (RS 26–100) [7–9]. An analysis of histologic subtypes was not included as part of the seminal validation studies, leaving some lack of clarity regarding the applicability of this molecular platform in estimating the risk of recurrence in lobular breast cancer.

Agnostic of histological subtype, decision impact studies, to date, report up to a 30–60% reduction in adjuvant chemotherapy administration with the incorporation of RS testing [9–12] The objectives of this study were to assess the change in chemotherapy recommendations in ILC with the use of RS and to assess the economic impact of this assay use in the publicly funded Irish healthcare system. We examined patterns of recurrence score and chemotherapy use in this Irish population of patients with ILC using real-world data and calculated chemotherapy costs. We also reviewed associations between grade, oncotype score, and age.

## 2. Materials and Methods

Between October 2011 and February 2019, a retrospective, cross-sectional observational study was conducted in Ireland. Patients with a diagnosis of HR+ early-stage breast cancer whose tumour specimens had 21-gene testing performed were identified. From October 2011 to February 2013, patients were identified from eight cancer centres nationally through the pathology departments at each of the cancer centres. These datasets were prospectively created as part of a national initiative to monitor the rollout of this technology. Between February 2013 and February 2019, patients were identified exclusively from the pathology department in St. Vincent's Healthcare Group. Data was not included from the other 7 national centres during this timeframe due to changes to recording and data-sharing principles. The patient population in St. Vincent's Healthcare Group is representative of the patient population nationally. From this large dataset, patients with ILC were extracted for analysis.

Data were collected from electronic patient records and paper medical notes using a standardised template. The results were manually recorded in the primary database. Patient characteristics recorded include age at diagnosis, tumour size, presence or absence of lymphovascular invasion (LVI), lymph node positivity, AJCC stage, and RS result. Treatment details for each patient were also gathered, including breast-conserving surgery (BCS) or mastectomy, if adjuvant treatment was received, and the recommendations post-RS testing. Chemotherapy details recorded included regimen type and number of cycles. Patients ≤50 years old were stratified into low RS (0–15), intermediate RS [13–22], and high RS (>25) risk groups based on Tailor X groupings [8].

Clinical risk was also calculated for each patient; low clinical risk was defined as a 92% probability at 10 years of breast cancer specific survival with adjuvant endocrine therapy, without systemic chemotherapy [23]. This is based on the work of the Early Breast Cancer Trialists' Collaborative Group meta-analysis, which has been validated in several datasets [23,24]. Low clinical risk was defined as grade 1 (G1) tumour ≤ 3 cm, grade 2 (G2) tumour ≤ 2 cm, or grade 3 (G3) tumour ≤ 1 cm [9]. Clinical risk was deemed to be high if the low-risk criteria were not met. All data were de-identified for patient confidentiality, and standardised with uniform nomenclature for statistical analysis. Datalock was December 2019. The outcomes measured were the change in treatment recommendation for each patient following RS testing and the net percentage reduction in chemotherapy use as a result. The net percentage reduction in chemotherapy is calculated as follows: those who received chemotherapy as a percentage of those who would otherwise have received chemotherapy if RS testing was not used.

### 2.1. Statistical Analysis

Descriptive statistics were used to analyse the data. Mean and standard deviation were reported for continuous variables, frequency, and proportion for categorical variables. MedCalc (Version 19) was the statistical software used.

### 2.2. Decision Impact Analysis

To quantify the impact of the assay, a pre-test treatment decision was determined for each patient in the dataset. This was based on the results of a survey of breast medical oncologists working in Ireland to generate a consensus opinion. In the absence of RS testing, patients with a G1 tumour would not receive chemotherapy, and those with a G2/3 tumour would receive chemotherapy. The details of this survey have been reported previously [10]. The change in treatment recommendation following RS testing was calculated in addition to the net percentage reduction in chemotherapy use. A separate analysis was undertaken assessing the impact of clinical risk combined with the recurrence score in identifying those patients ≤50 years of age with high clinical risk who are potentially most likely to derive benefit from adjuvant chemotherapy.

### 2.3. Budget Impact Analysis

An analysis of the overall costs was calculated using a simple budget impact model. The base currency is Euro (EUR), and costs are calculated in this currency over the time horizon of the study, taking a cost point for each element (chemotherapy costs, GCSF, and chemotherapy adverse events costs) within the time period of the study. The cost of the assay is static over time. Discounting is not required. The patient population is the number of patients projected to avail of this technology. Based on the net percentage reduction of adjuvant chemotherapy use as a result of the 21-gene testing, the cost of chemotherapy before and after testing was calculated over the time horizon of the study. This time period was chosen to reflect a slightly longer budgetary cycle, correlating with the rollout of this new technology, and in order to more accurately project the budgetary impact of the introduction. A third-party payer perspective was used for the analysis, reflecting the

national healthcare funding structure. The cost of the RS test was incorporated into the formula to calculate the overall net savings as per the equation below.

$$([\text{\#patients}] \times [\text{cost per 21-gene test}]) - ([\text{net chemotherapy reduction}] \times [\text{\#patients}] \times [\text{cost of chemotherapy}])$$

*2.4. Costs*

Cost data for chemotherapy administration in the hospital daycare setting were obtained from the National Healthcare Pricing Regulatory Authority [11]. This cost data includes chemotherapy drug costs, pharmacy compounding costs, and nursing and administrative costs associated with chemotherapy delivery in the oncology dayward setting, standardised across nationally approved chemotherapy regimens. The average chemotherapy cost per patient was calculated based on a weighted average of the chemotherapy regimen prices according to the proportion of use of each regimen within the dataset. In addition, the costs for granulocyte-colony-stimulating factor (G-CSF) treatment were obtained from the national Health Products Regulatory Authority's (HPRA) high-tech medicines published price list for 2019, priced at €976.78 [12]. The 21-gene assay price of €3180 per test was obtained from the manufacturer's listed price. The costs of managing adverse events related to chemotherapy administration were based on a national pharmacoeconomic assessment [12]. The total cost of chemotherapy per patient, in EUR, was calculated at €7313.84 (see Table 1).

**Table 1.** Chemotherapy cost calculation.

| Parameter | Cost |
|---|---|
| Chemotherapy incl. administration costs per regimen | €2380.84 |
| G-CSF x4 cycles | €4177.52 |
| Adverse events | €756.00 |
| Total | €7313.84 |

## 3. Results

Between October 2011 and February 2019, 168 patients with node-negative ILC were identified for analysis. The mean age was 58.67 years (SD 8.11) and was consistent across all recurrence score groups. The mean tumour size was 2.3 cm (range 0.7–5.8 cm). Twenty-nine (17%) patients were aged $\leq 50$ years at diagnosis, compared to 139 (83%) aged >50 years. Patient and tumour characteristics are summarised in Table 2.

The majority of patients (*n* = 154, 92%) had G2 tumours, of which, 141 (92%) had a RS $\leq$ 25, and 13 (8%) had RS > 25. Seven patients (4%) had G1 disease, all of whom had a RS $\leq$ 25, and 7 patients (4%) had G3 disease, all with a RS $\leq$ 25. On analysis of the recurrence score within the total group, 155 patients (92.3%) had a low RS ($\leq$25), 12 (7.1%) had a high RS (>25), and 1 (0.6%) had an unknown RS. Of the 29 patients aged $\leq$50 years at diagnosis, 1 (3.5%), 27 (93%), and 1 (3.5%) had G1, G2, and G3 disease, respectively. Furthermore, 16 patients aged $\leq$50 years (55%) had RS $\leq$ 15, 6 (21%) had RS 16–20, 5 (17%) had RS 21–25, 1 (3.5%) had RS > 25, and 1 patient (3.5%) had unknown RS. Of the total patient group, all were ER+, and 70 (42%) were PR+. Thirty-eight patients (23%) had a tumour size of 3 cm or above.

The majority of patients underwent breast-conserving surgery (*n* = 126, 75%), with the remainder undergoing mastectomy. Eighty patients (47.6%) had AJCC stage I disease, of which 73 (91.2%) had RS $\leq$ 25. Seventy-two patients (43%) had stage IIA disease, of which 5 (6.9%) had a RS > 25. Two patients (1.1%) had stage IIB disease, of which 1 (50%) had a RS of 0–15, and 1 (50%) had a RS of 21–25. In total, 118 patients (70%) underwent radiotherapy.

**Table 2.** Clinical characteristics and treatment received by RS.

|  | Total (*n*) | RS 0–15 (%) | RS 16–20 (%) | RS 21–25 (%) | RS 26–100 (%) | Missing RS (%) |
|---|---|---|---|---|---|---|
| LN0 | 168 | 76 (45.2) | 46 (27.4) | 33 (19.6) | 12 (7.2) | 1 (0.6) |
| Mean age [SD] | 59 [8.11] | 57 | 60 | 60 | 61 | 48 |
| Age < 50 | 29 | 16 (55.0) | 6 (21.0) | 5 (17.0) | 1 (3.5) | 1 (3.5) |
| Age > 50 | 139 | 60 (43.2) | 40 (28.8) | 28 (20.1) | 11 (7.9) | 0 (0.0) |
| LVI+ | 11 | 6 (54.5) | 2 (18.2) | 3 (27.3) | 0 (0.0) | 0 (0.0) |
| LVI- | 101 | 46 (45.4) | 29 (28.7) | 17 (16.8) | 8 (7.9) | 1 (1.1) |
| LVI Unknown | 56 | 24 (42.9) | 15 (26.8) | 13 (23.2) | 4 (7.1) | 0 (0.0) |
| Stage IA | 80 | 36 (45.0) | 24 (30.0) | 13 (16.2) | 6 (7.5) | 1 (1.3) |
| Stage IIA | 72 | 31 (43.1) | 19 (26.4) | 17 (23.6) | 5 (6.9) | 0 (0.0) |
| Stage IIB | 2 | 1 (50.0) | 0 (0.0) | 1 (50.0) | 0 (0.0) | 0 (0.0) |
| Unknown | 14 | 8 (57.1) | 3 (21.4) | 2 (14.4) | 1 (7.1) | 0 (0.0) |
| Grade 1 | 7 | 3 (42.9) | 4 (57.1) | 0 (0.0) | 0 (0.0) | 0 (0.0) |
| Grade 2 | 154 | 68 (44.8) | 39 (26.0) | 32 (20.8) | 12 (7.8) | 1 (0.6) |
| Grade 3 | 7 | 4 (57.1) | 2 (28.6) | 1 (14.3) | 0 (0.0) | 0 (0.0) |
| BCS | 126 | 55 (43.7) | 34 (27.0) | 28 (22.2) | 9 (7.1) | 0 (0.0) |
| Mastectomy | 42 | 21 (50.0) | 12 (28.6) | 5 (11.9) | 3 (7.1) | 1 (2.4) |
| Radiotherapy | 118 | 53 | 5 | 29 | 5 | 0 |
| CT received | 35 | 3 (8.6) | 4 (11.4) | 15 (42.8) | 12 (34.3) | 1 (2.9) |

BCS = breast conserving surgery. RS = recurrence score. CT = chemotherapy.

Without RS testing, 161 patients were recommended adjuvant chemotherapy on the basis of G2 or G3 pathology, in line with our pre-test survey. Post-RS testing, 126 patients (78%) had a change in chemotherapy decision, with all changing from chemotherapy to hormone therapy alone. In total, 35 patients (22%) received chemotherapy. Of those patients treated with chemotherapy, 3 (9%) had RS 0–15, 19 (54%) had RS 16–25, 12 (34%) had RS > 25, and 1 (3%) had unknown RS. The most common regimen the patients received was docetaxel plus cyclophosphamide (*n* = 28, 80%).

*3.1. Decision Impact Analysis*

3.1.1. Pre-21-Gene Test Chemotherapy Recommendation

One hundred and sixty-one patients (96%) in this group had G2 or G3 tumours. Based on this, they were assumed to have a positive pre-21-gene test recommendation for adjuvant chemotherapy in addition to hormone therapy. The remaining 7 patients with G1 disease were assumed to have a negative pre-test recommendation (i.e., they would not have received adjuvant chemotherapy in addition to hormone therapy). Of the 161 patients initially recommended adjuvant chemotherapy pre-test, only 35 patients received chemotherapy post-21-gene testing, representing a net 78% reduction in chemotherapy use.

3.1.2. Post 21 Gene Test Chemotherapy Recommendation

The use of the 21-gene assay achieved a 78% change in treatment decision (*n* = 126). For all of these patients, the change in treatment was from chemotherapy to no chemotherapy being recommended in addition to adjuvant hormone therapy. In patients with G2 disease (*n* = 54), 77% had a change in chemotherapy recommendation in favour of no adjuvant chemotherapy. Among patients with G3 disease (*n* = 7), all 7 (100%) had a change in adjuvant treatment recommendation in favour of no chemotherapy. None of the 7 patients with G1 disease experienced a change in treatment recommendation based on the use of the 21-gene assay.

Of the 126 patients who had a change in chemotherapy recommendation, 70 (56%) had RS 0–15 and 56 (44%) had RS 16–25. Among patients with a RS ≤ 25 (*n* = 155), the recommendation for adjuvant chemotherapy was reduced from 148 (95%) to 22 patients (14%). Among patients with a RS > 25 (*n* = 12), there was no change in treatment decision. Table 3 shows the distribution of RS among different subgroups for the entire study population (*n* = 168).

**Table 3.** RS distribution in study population.

|  | Total | RS 0–15 | RS 16–25 | RS 26–100 | Missing RS |
|---|---|---|---|---|---|
| Pre Test Yes | 161 | 73 | 75 | 12 | 1 |
| Pre Test No | 7 | 3 | 4 | 0 | 0 |
| Post Test Yes | 35 | 3 | 19 | 12 | 1 |
| Post Test No | 133 | 73 | 60 | 0 | 0 |

### 3.2. Decision Impact in the ≤50 Age Group

Among the 29 patients aged ≤50 years, 16 (55%) had a RS 0–15, 6 (21%) 16–20, 5 (17%) 21–25, and 1 (3.5%) had a RS > 25. There was 1 patient (3.5%) with unknown RS in this group. Twenty-seven patients (93%) had G2 disease, 1 patient had G1, and 1 patient had G3 disease. Among the 16 patients with RS 0–15, 2 (13%) received adjuvant chemotherapy. One patient with a recurrence score of 16–20 received chemotherapy, three patients with a score of 21–25, and one patient with a score of 26–100 also received adjuvant chemotherapy. All seven patients had G2 disease. Without the use of RS testing, 28 patients (97%) in this group of patients aged ≤50 would have been recommended adjuvant chemotherapy based on G2 or G3 disease. With the incorporation of this assay, 8 patients (28%) received a chemotherapy recommendation equating to a 71% reduction in chemotherapy use in patients aged ≤50 (see Table 4).

**Table 4.** Patients aged ≤50 years: age, grade (G), recurrence score (RS), chemotherapy (CT).

|  | Total | RS 0–15 | RS 16–20 | RS 21–25 | RS 26–100 | Missing |
|---|---|---|---|---|---|---|
| 40–50 | 28 | 15 | 6 | 5 | 1 | 1 |
| 30–39 | 1 | 1 | 0 | 0 | 0 | 0 |
| G1 | 1 | 1 | 0 | 0 | 0 | 0 |
| G2 | 27 | 14 | 6 | 5 | 1 | 1 |
| G3 | 1 | 1 | 0 | 0 | 0 | 0 |
| CT | 8 | 2 | 1 | 3 | 1 | 1 |

### 3.3. Decision Impact in the ≤50 Age Group per Clinical Risk

We examined the potential role of clinical risk in our group of younger patients aged ≤50 years. Data on tumour grade was missing for one patient, rendering calculation of clinical risk impossible. Twelve (41%) of the patients aged ≤50 years had a high clinical risk based on the criteria outlined previously (low clinical risk was defined as grade 1 (G1) tumour ≤ 3 cm, grade 2 (G2) tumour ≤ 2 cm, or grade 3 (G3) tumour ≤ 1 cm [9]). Clinical risk was deemed high if the low-risk criteria were not met. Of these 12 patients, 6 had RS 16–20. One of these 6 patients received adjuvant chemotherapy. The remaining 17 patients (59%) had low clinical risk.

### 3.4. Budget Impact Analysis

The net reduction of 75% in the use of adjuvant chemotherapy resulted in savings of €921,543.84 in treatment costs. Without the use of the RS, the cost of chemotherapy for

161 patients who had a pre-test recommendation for chemotherapy was €1,177,528.24. Incorporating the assay cost for all 168 patients, the net savings achieved totalled €387,283.84.

## 4. Discussion

This study provides a real-world analysis of the clinical and economic impact of 21-gene testing in 168 women with lymph node-negative, invasive, lobular, early-stage breast cancer in Ireland. Paik et al., demonstrated a 4.4% absolute benefit of chemotherapy with endocrine therapy in patients with RS above 31 at 10-year disease-free recurrence [6]. The seminal TAILOR-X prospective trial of over 10,000 women with node-negative, HR+, and HER2- breast cancer demonstrated the utility of incorporating a RS into the patient's treatment paradigm in order to identify those who are unlikely to benefit from adjuvant chemotherapy in addition to endocrine therapy. The RS groups are low- and intermediate-risk below 26 and a high RS of 26 and above. The use of RS testing with the modified RS groups resulted in a 75% reduction in chemotherapy administration in this study. In our previously published work analysing a larger dataset with predominantly invasive ductal carcinoma, we found a 62.5% change in chemotherapy recommendation [13]. The TAILOR-x trial confirmed that patients aged >50 years with a RS ≤ 25 do not benefit from the addition of chemotherapy to endocrine therapy and may be effectively treated with ET alone.

The homogeneity typically seen in ILC presents a challenge when deciphering the role of molecular testing with respect to informing adjuvant chemotherapy decisions. Despite the differences in clinical, pathological, and genomic characteristics of ILC compared to IDC, most studies have not separated out the histological subtypes when assessing the role of adjuvant chemotherapy [2]. ILC represents a distinct subset of early-stage breast cancer that is often characterised by a lack of E-cadherin protein expression, ER positivity, and HER2 negativity [14]. A number of studies have explored the molecular portrait of ILC. Ciriello et al. found that mutations targeting CDH1, PIK3CA, RUNX1, TBX3, and FOXA1 are more prevalent in ILC tumours [15]. Desmedt and colleagues identified alterations in one of three key genes of the PI3K pathway: PIK3CA, PTEN, and AKT1, in 50% of ILC breast cancer cases, each more frequently mutated in ER-positive/HER2-negative ILC breast cancer than in ER-positive/HER2-negative invasive ductal breast cancer tumours [16]. While this knowledge reveals an increasingly distinct tumour landscape, treatment paradigms tailored to ILC lag behind.

We found that most patients (92%) had a recurrence score of <25. This finding is consistent with other studies evaluating the utility of oncotype in ILC, where the majority have RS < 25 [17,18]. Tsai et al., used tumour characteristics such as size, grade, ER and PR positivity, and Ki-67 proliferation rate to distinguish between low versus elevated recurrence risk in patients with ILC [19]. Interestingly, in this study, PR positivity expression was the most important factor shown to distinguish lower- and more elevated-risk tumours. The Ki-67 proliferating marker was also shown to distinguish low- and elevated-risk tumours. One limitation of Ki-67 is that it is not universally reported on breast cancer samples. Conlon and colleagues showed that ILC morphology (classical vs. pleomorphic) cannot be used as a surrogate for RS, as both ILC subtypes demonstrate a RS that spans more than one risk category [18]. Further research is needed to define reliable predictive and prognostic risk characteristics in ILC as the role of molecular platforms in this tumour setting evolves.

Similar to several studies, this study showed that patients with ILC are, on average, in older age groups (>50 years) (82%, *n* = 139) at presentation. This older age at diagnosis could be due to a low ILC proliferative rate or the diminished utility of diagnostic tools in ILC [20]. Among the 29 women aged ≤50 years in our study, we found a 71% reduction in adjuvant chemotherapy use, with only 8 patients (23%) ultimately receiving a chemotherapy recommendation. Marmor et al., looked at the survival of ILC compared to IDC with adjuvant chemotherapy [2]. The 10-year overall survival rate was 94% with endocrine therapy alone vs. 92% with endocrine therapy plus chemotherapy. They did not observe

an improvement in overall survival for ILC. As the majority of ILCs are hormone receptor-positive and have poor chemo-sensitivity, endocrine-based therapy is often favoured as a therapeutic option [21]. Previous studies analysed the efficacy of adjuvant letrozole compared to tamoxifen. The disease-free survival rate has been reported at 66% with tamoxifen versus 82% with letrozole-treated ILC [21,22]. Despite the increased response to endocrine therapy, there are still treatment challenges due to endocrine resistance. Ongoing studies, such as the ROLO study assessing crizotinib and fulvestrant in E-positive lobular breast cancer, are addressing the role of targeted therapies in this disease subtype [22].

It is important to note that the budget impact analysis presented here is from the perspective of the exchequer and, therefore, does not take into account, in a monetary sense, the personal financial impact on the patient of avoiding or receiving chemotherapy. This additional impact may include elements such as revenue lost from leaving the workforce (temporarily or permanently) and expenses such as transport costs, hospital parking costs, childcare costs, and the cost of supportive medications.

### 5. Conclusions

In this study, we estimated that using 21-gene testing in patients with ILC, with the incorporation of chemotherapy costs, led to a net savings of €387,283.84 to the Irish health service. With ILC representing 10–15% of invasive breast cancers, the overall impact, firstly on treatment decisions and avoidance of adverse treatment effects on our patients and secondly on the Irish economy, is considerable. While there is some literature focusing on the role of the 21-gene test in the management of ILC, this study adds to that existing literature, thus increasing the available data on RS results in ILC in a real-world setting. Further research is needed to more fully elucidate the role of chemotherapy or targeted therapies and the optimal role of genomic testing in this context.

**Author Contributions:** Study concept and design: C.M.Q. and J.M.W. Acquisition of data: L.M.M., M.T., F.A.R., F.K., J.P.C., D.E., J.G.G., J.R., D.P.M., M.G., M.O., D.O., M.K., M.J.K. and S.O. Drafting, reviewing, and critical analysis of data, manuscript, and tables: L.M.M., M.T., S.J.M., C.M.Q., R.S.P. and J.M.W. Final approval of version to be published and oversight for accuracy of the work: L.M.M., M.T., F.A.R., F.K., D.E., J.G.G., J.R., D.P.M., M.G., M.O., D.O., M.K., M.J.K., S.O., S.J.M., J.P.C., C.M.K., R.S.P., C.M.Q. and J.M.W. All authors have read and agreed to the published version of the manuscript.

**Funding:** This research received no external funding.

**Institutional Review Board Statement:** This retrospective review of patient data did not require ethical approval in accordance with local/national guidelines. The study is registered with and approved by the Clinical Audit Committee at St Vincent's University Hospital, Dublin 4, Ireland. This research was conducted in accordance with the 1964 Helsinki Declaration and its later amendments, and national and institutional research standards.

**Informed Consent Statement:** Written informed consent from participants was not required in accordance with local/national guidelines.

**Data Availability Statement:** All data analysed during this study are included in this article. The fully anonymised datasets generated and analysed during the current study are available from the corresponding author on reasonable request.

**Conflicts of Interest:** Author Steve Millen was employed by the company Exact Sciences Ltd. The remaining authors declare that the research was conducted in the absence of any commercial or financial relationships that could be construed as a potential conflict of interest.

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
