# Peer review of "Real-World Analysis of the Clinical and Economic Impact of the 21-Gene Recurrence Score (RS) in Invasive Lobular Early-Stage Breast Carcinoma in Ireland"

_curroncol, doi:10.3390/curroncol31030098_

Round 1

Reviewer 1 Report

Comments and Suggestions for Authors

This is a well-written, compact paper with a clearly stated research question and an analysis that appropriately answers that question. My only comment is whether the cost savings identified might be a lower bound. The benefits of the testing are limited to the costs avoided from not providing chemotherapy. Are there any other patient benefits that are not being considered, such as reduced incidence of side effects from chemotherapy, treatment-related illness, time off work, etc?

Author Response

Thank you for your time and expertise in reviewing this paper. Yes, I agree there are other factors not relating directly to chemotherapy costs which would be relevant here. I have incorporated comments to expand on that, although it is not directly included in a 3rd party payer budget impact. I agree the cost savings are likely a lower bound. Many thanks again. 

Reviewer 2 Report

Comments and Suggestions for Authors

This is a study that attempts to examine the clinical and economic impact of the use of the 21-gene recurrence score among patients with invasive lobular early-3 stage breast carcinoma in Ireland. However, I have the following concerns.

Methods:
Why were there two different selection process - eight cancer centres nationally through their pathology departments from Oct 2011 to Feb 2013 vs. pathology department in St Vincent’s Healthcare group from Feb 2013 to Feb 2019? Are the patients from the St Vincent’s Healthcare group representative of the national population?

How was relative risk computed, was it via regression?
What statistical software was used?

Can the authors please summarise in a table or figure what exactly the treatments for each patient group are because it's currently unclear what they are and so it's difficult to see what would have changed in the Results section?

Are the outcomes measured here "change in treatment recommendation following RS testing" and "net percentage reduction in chemotherapy use"? If so, can the authors state it as so for clarity please?
Are all the outcomes measured via both the survey and the TAILOR-x trial? If not, which measure was used to measure which outcome?
Can there be more description about the TAILOR-x trial please?

How exactly was the impact of clinical risk combined with the recurrence score in identifying those patients ≤50 with high clinical risk assessed?

What is the time horizon of the budget impact analysis (BIA)?

What is the base currency used?

How was the number of patients determined in the BIA?

How was the net percentage reduction of adjuvant chemotherapy use computed?

Please be consistent with the terminology, chemotherapy was sometimes abbreviated as CT and spelt out as chemotherapy during other times.

Results:
Missing relative risk in the Results section.

Were there ever any mean, median and standard deviation reported in this study? Because the Methods suggested so.

Does Table A2 refer to the number of patients? The main text describing Table A2 from lines 200-204 did not tally with the values in the Table. For clarity, please either revise Table A2 or state how the numbers in the main text were derived.

Discussion:
What are the limitations of this study?

How would the specificity and sensitivity of the assay affect the results?

Author Response

Many thanks for your time and expertise in reviewing this paper. Please find below comments in relation to the points raised. I have addressed and revised the paper to incorporate all suggested edits. Thanks again. 

Why were there two different selection process - eight cancer centres nationally through their pathology departments from Oct 2011 to Feb 2013 vs. pathology department in St Vincent’s Healthcare group from Feb 2013 to Feb 2019? -reflecting different recording procedures in operation. For data consistency, after 2013, patient records were not maintained in the same way in the other cancer centres so could not be included. Are the patients from the St Vincent’s Healthcare group representative of the national population? -Yes. 

How was relative risk computed, was it via regression? -edited and removed. Numbers too small
What statistical software was used? MedCalc

Can the authors please summarise in a table or figure what exactly the treatments for each patient group are because it's currently unclear what they are and so it's difficult to see what would have changed in the Results section? Edited.Thankyou.

Are the outcomes measured here "change in treatment recommendation following RS testing" and "net percentage reduction in chemotherapy use"? If so, can the authors state it as so for clarity please? -edited for clarity
Are all the outcomes measured via both the survey and the TAILOR-x trial? If not, which measure was used to measure which outcome? -the survey. Edited for clarity
Can there be more description about the TAILOR-x trial please? -included,thank you. 

How exactly was the impact of clinical risk combined with the recurrence score in identifying those patients ≤50 with high clinical risk assessed? -additional description for clarity

What is the time horizon of the budget impact analysis (BIA)? -edited.2011-2019

What is the base currency used? -euro

How was the number of patients determined in the BIA? edited for clarity.

How was the net percentage reduction of adjuvant chemotherapy use computed? -simple computation. Those who received chemo as a percentage of those who would otherwise have. 

Please be consistent with the terminology, chemotherapy was sometimes abbreviated as CT and spelt out as chemotherapy during other times. -changed for consistency. 

Results:
Missing relative risk in the Results section. -edited

Were there ever any mean, median and standard deviation reported in this study? Because the Methods suggested so. -edited

Does Table A2 refer to the number of patients? The main text describing Table A2 from lines 200-204 did not tally with the values in the Table. For clarity, please either revise Table A2 or state how the numbers in the main text were derived. -edited for clarity

Discussion:
What are the limitations of this study? -discussed

How would the specificity and sensitivity of the assay affect the results?  -edited. 

Reviewer 3 Report

Comments and Suggestions for Authors

Excellent paper-only a very few minor suggestions:

1. Line 53- I advise adding the word often to " ILC often has many initially favourable prognostic factors".  I don't think this applies to every ILC

2.Lines 64-67 you state:" By better selecting those patients likely to benefit from  chemotherapy, we ensure patients avoid the short- and long-term toxicities of cytotoxic  treatment, such as nausea, alopecia, early menopause, infertility, and the psychological  impact of these effects. "

I suggest rewording this to something like -

By better selecting those patients likely to benefit from  chemotherapy, fewer patients will be exposed to chemotherapy and the potential short- and long-term toxicities of cytotoxic  treatment, such as nausea, alopecia, early menopause, infertility, and the psychological  impact of these effects. "

3. Lines 242-244- I suggest you change to : ILC represents a distinct subset of ESBC which is often characterised by lack of E-Cadherin protein expression, ER positivity, and 243 HER 2 negativity

That concludes my very minor suggestions of this very well-written paper.

Author Response

Many thanks for your time and expertise in reviewing this paper. I have incorporated all of your suggestions into the revised document submitted today. Thanks again. 

  1. Line 53- I advise adding the word often to " ILC often has many initially favourable prognostic factors".  I don't think this applies to every ILC

2.Lines 64-67 you state:" By better selecting those patients likely to benefit from  chemotherapy, we ensure patients avoid the short- and long-term toxicities of cytotoxic  treatment, such as nausea, alopecia, early menopause, infertility, and the psychological  impact of these effects. "

I suggest rewording this to something like -

By better selecting those patients likely to benefit from  chemotherapy, fewer patients will be exposed to chemotherapy and the potential short- and long-term toxicities of cytotoxic  treatment, such as nausea, alopecia, early menopause, infertility, and the psychological  impact of these effects. "

3. Lines 242-244- I suggest you change to : ILC represents a distinct subset of ESBC which is often characterised by lack of E-Cadherin protein expression, ER positivity, and 243 HER 2 negativity

Round 2

Reviewer 2 Report

Comments and Suggestions for Authors

In the authors' response, please indicate where the changes are because it's difficult to tell what the changes are.
I couldn't locate the changes to the following to assess whether sufficient edits were made:
1. "Can the authors please summarise in a table or figure what exactly the treatments for each patient group are because it's currently unclear what they are and so it's difficult to see what would have changed in the Results section? Edited.Thankyou."?
2. "Can there be more description about the TAILOR-x trial please? -included,thank you."
More specifically, is the risk stratification in the following line based on the TAILOR-X trial?
"Patients ≤50 years old were stratified into low RS (0-15), intermediate RS (16-25), and high RS (>25) risk groups".
If so, please explicitly write that this is based on the TAILOR-X trial in order to tie the information together.
3. "How exactly was the impact of clinical risk combined with the recurrence score in identifying those patients ≤50 with high clinical risk assessed? -additional description for clarity"
What is the separate analysis?
4. "How was the number of patients determined in the BIA? edited for clarity."
5. "How would the specificity and sensitivity of the assay affect the results?  -edited. "

The outcomes were measured using survey but was mentioned with the data collected from the electronic patient records. Therefore implying they were collected using electronic patient records since there was no explicit statement regarding how the outcomes were collected. Please take note of the flow of the information and ensure that similar information are grouped together instead of having it in bits and pieces.

Why is the time horizon of 2011 to 2019 chosen for the BIA?
If this time horizon does not coincide with the budget cycle, please explain why it goes against the guideline by the Budget Impact Analysis Good Practice II Task Force (Sullivan et al, 2014) of having BIA "presented for the time horizons of most relevance to the budget holder" or consider redoing your analysis.

Please also include the year of currency with the base currency.

"How was the net percentage reduction of adjuvant chemotherapy use computed? -simple computation. Those who received chemo as a percentage of those who would otherwise have. "
Please have the explanation included in the main text.

Table A1: The mean age does not contain any SD

Author Response

In the authors' response, please indicate where the changes are because it's difficult to tell what the changes are.
I couldn't locate the changes to the following to assess whether sufficient edits were made:
1. "Can the authors please summarise in a table or figure what exactly the treatments for each patient group are because it's currently unclear what they are and so it's difficult to see what would have changed in the Results section? Edited.Thankyou."? -The authors feel this is adequately presented; patients with grade 2/3 tumours would have (hypothetically) received chemotherapy pre the introduction of RS testing. Please note this study was conducted after the introduction of RS testing so very few of those G2/3 patients actually received chemotherapy.

  1. "Can there be more description about the TAILOR-x trial please? -included,thank you."
    More specifically, is the risk stratification in the following line based on the TAILOR-X trial?
    "Patients ≤50 years old were stratified into low RS (0-15), intermediate RS (16-25), and high RS (>25) risk groups".
    If so, please explicitly write that this is based on the TAILOR-X trial in order to tie the information together. -see line 110

  2. "How exactly was the impact of clinical risk combined with the recurrence score in identifying those patients ≤50 with high clinical risk assessed? -additional description for clarity"
    What is the separate analysis? -I have included a separate heading to highlight this paragraph. See line 237 of the manuscript.

  3. "How was the number of patients determined in the BIA? edited for clarity." -based on the projected number of patients availing of this technology (RS test)

  4. "How would the specificity and sensitivity of the assay affect the results?  -edited. "

    The outcomes were measured using survey but was mentioned with the data collected from the electronic patient records. Therefore implying they were collected using electronic patient records since there was no explicit statement regarding how the outcomes were collected. Please take note of the flow of the information and ensure that similar information are grouped together instead of having it in bits and pieces. -the outcomes were not measured using the survey. The only purpose of the survey was to establish the assumption for the decision impact analysis, ie. oncologists nationally were surveyed regarding treatment; whether or not they would give adjuvant chemotherapy to a patient with grade 1, 2, or 3 breast cancer. See lines 128-133

    Why is the time horizon of 2011 to 2019 chosen for the BIA?
    If this time horizon does not coincide with the budget cycle, please explain why it goes against the guideline by the Budget Impact Analysis Good Practice II Task Force (Sullivan et al, 2014) of having BIA "presented for the time horizons of most relevance to the budget holder" or consider redoing your analysis. -line 141-144

    Please also include the year of currency with the base currency. -see line 140

    "How was the net percentage reduction of adjuvant chemotherapy use computed? -simple computation. Those who received chemo as a percentage of those who would otherwise have. "
    Please have the explanation included in the main text. -see line 122, 123

    Table A1: The mean age does not contain any SD -added to the table

Reviewer 3 Report

Comments and Suggestions for Authors

Thank you for addressing my suggestions

Author Response

Thank you

Round 3

Reviewer 2 Report

Comments and Suggestions for Authors

The following has not been addressed adequately.

"I have included a separate heading to highlight this paragraph. See line 237 of the manuscript."
Methods should be in the Methods section and not in the Results section.

"based on the projected number of patients availing of this technology (RS test)"
But I didn't see this written in the main text. Anticipated uptake of new intervention is important in BIA and having it written removes assumption of what the "number of patients" the authors are talking about.

"How would the specificity and sensitivity of the assay affect the results? -edited."
I don't see this being addressed.

"costs are calculated in this currency in the year that they occurred"
Can the authors explain why the costs were not adjusted to a current price (e.g. 2019 Euro)? Just in case I wasn't clear, I'm not talking about discounting.

Author Response

"I have included a separate heading to highlight this paragraph. See line 237 of the manuscript."
Methods should be in the Methods section and not in the Results section.

-Thank you. It is also included in the methods section. See paragraph beginning at line 113.

"based on the projected number of patients availing of this technology (RS test)"
But I didn't see this written in the main text. Anticipated uptake of new intervention is important in BIA and having it written removes assumption of what the "number of patients" the authors are talking about.

-This detail has been added to the manuscript for clarity. See line 142, 143

"How would the specificity and sensitivity of the assay affect the results?"
I don't see this being addressed -please see attached word document for a detailed explanation around this question. 

Can the authors explain why the costs were not adjusted to a current price (e.g. 2019 Euro)? Just in case I wasn't clear, I'm not talking about discounting.

-thank you. The costs used (chemotherapy, GCSF, and adverse chemo-related events) are all from a cost point within the study period and do not require further adjustment. The cost of the assay is static over time. We feel this is correctly presented. 

Round 4

Reviewer 2 Report

Comments and Suggestions for Authors

Noted with thanks regarding the responses.

However, one of the responses was "The costs used (chemotherapy, GCSF, and adverse chemo-related events) are all from a cost point within the study period and do not require further adjustment. The cost of the assay is static over time. We feel this is correctly presented." This is different from what's been revised in the manuscript, "...and costs are calculated in this currency adjusted for price changes over the time horizon of the study".

Since there is no adjustment of costs, please state in the manuscript that there was no cost adjustment and the reason(s) for not having any cost adjustments.

Author Response

Thank you again for your comments. The change has been made as suggested.